# Valence-isomer selective cycloaddition reaction of cycloheptatrienes-norcaradienes

Shingo Harada [1] ✉, Hiroki Takenaka[1,2], Tsubasa Ito[1,2], Haruki Kanda[1] & Tetsuhiro Nemoto [1] ✉

The rapid and precise creation of complex molecules while controlling multiple selectivities is the principal objective in synthetic chemistry. Combining data science and organic synthesis to achieve this goal is an emerging trend, but few examples of successful reaction designs are reported. We develop an artificial neural network regression model using bond orbital data to predict chemical reactivities. Actual experimental verification confirms cycloheptatriene-selective [6 + 2]-cycloaddition utilizing nitroso compounds and norcaradiene-selective [4 + 2]-cycloaddition reactions employing benzynes. Additionally, a one-pot asymmetric synthesis is achieved by telescoping the enantioselective dearomatization of non-activated benzenes and cycloadditions. Computational studies provide a rational explanation for the seemingly anomalous occurrence of thermally prohibited suprafacial [6 + 2]-cycloaddition without photoirradiation.

Cycloaddition reactions are universal molecular transformations for creating two σ-bonds to directly synthesize valuable (hetero)cyclic ring systems[1]. In general, conjugated olefins and unsaturated compounds are used as readily available reaction components for the powerful annulation strategy. An array of bioactive natural products and drug candidates has been naturally or artificially synthesized on the basis of pericyclic reactions[2,3]. Particularly noteworthy is the report by the Boger group on the rapid total synthesis of vindoline, the chemical precursor of an antineoplastic agent (vinblastine) via an intramolecular [4 + 2]/[3 + 2] cycloaddition cascade[4]. Antiviral Oseltamivir has also been synthesized by a stereoselective Diels-Alder reaction[5] for constructing functionalized core carbocycles[6,7]. Control of the site-[8], regio-[9,10]; and stereoselectivities[11,12] in intermolecular cycloadditions using structurally unsymmetrical reactants, however, remains challenging even in modern organic chemistry[13,14]. Cycloheptatriene (CHT) and norcaradiene (NCD) exist in a unique state of equilibrium as valence isomers, giving rise to a diverse range of reaction manifolds that lead to an array of similar, yet distinct, cycloadducts (Fig. 1a).

As part of our ongoing studies in chemical sciences, we developed a realm of dearomatization chemistry[15–17]. By taking advantage of a unique property of carbene species depending on the catalysis, chemoselective catalytic asymmetric dearomatization

(CADA) reactions[18–20] of phenols were achieved[16]. To further enhance the synthetic utility, a diazo-free method of generating silver-carbenes was developed[17,21] using a chiral counteranion strategy[22–26], which provided CHTs in an enantioenriched form (Fig. 1b)[27]. When the product reacted with 1,2,4-triazoline-3,5-dione as a dienophile, the Diels-Alder reaction proceeded by trapping NCDs. In contrast, other dienophile components were significantly less reactive, even with highly activated olefins. In the 1950s, Fukui proposed the frontier molecular orbital theory that molecular reactivity could be chemically described by the highest occupied and lowest unoccupied molecular orbitals (HOMO and LUMO, respectively)[28,29]. In the present day, quantum computations are widely feasible for estimating the underlying physical organic properties of object molecules. Our working hypothesis is that machine-learning algorithms of the frontier orbital energy gaps predicted by computations based on density functional theory (DFT) could find enophile candidates that react efficiently with CHT/ NCD[30–35]. By a combination of computations and machine learning with experimental validation (Fig. 1d)[36], reactions of CHTs with nitroso compounds were revealed to cause a remarkable suprafacial [$_\pi6_s$ + $_\pi2_s$]-cycloaddition that is thermally forbidden based on Woodward-Hoffmann rules[37–39]. Previous studies used nitroso compounds as dienophiles in [4 + 2]-cycloadditions[40–44], even in reactions

[1]Graduate School of Pharmaceutical Sciences, Chiba University, Chiba 260-8675, Japan. [2]These authors contributed equally: Hiroki Takenaka, Tsubasa Ito. ✉e-mail: Sharada@chiba-u.jp; tnemoto@faculty.chiba-u.jp

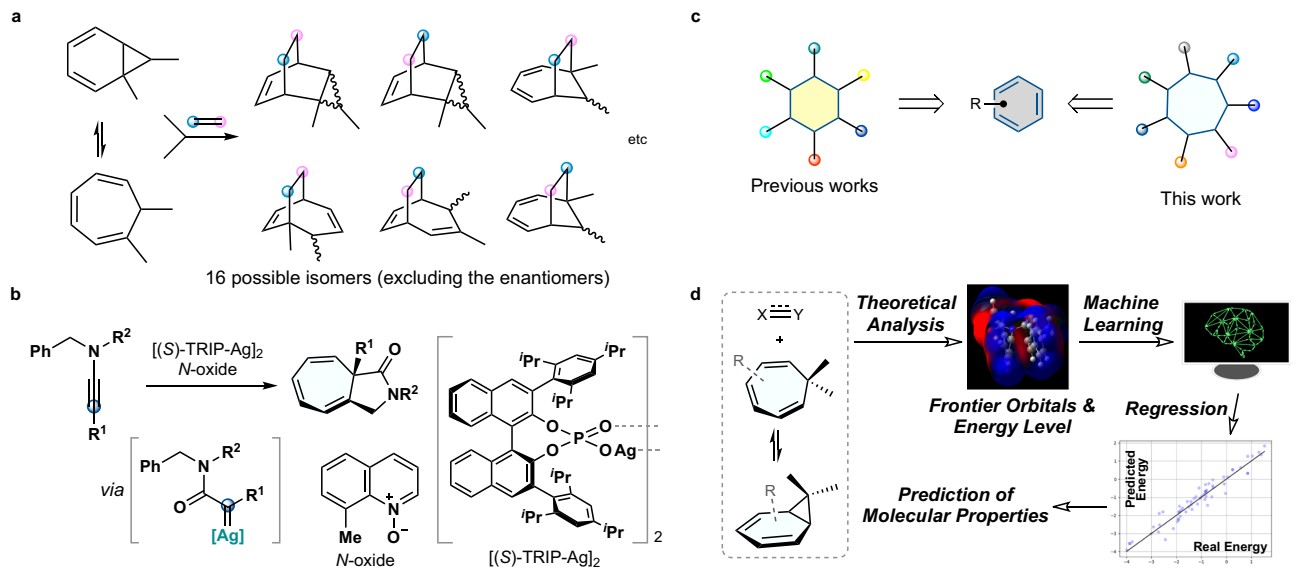

**Fig. 1 | Research background and strategies of the present work. a** Challenge of controlling the product selectivity during cycloaddition. **b** Carbene-mediated dearomatization of non-activated arenes. **c** Functionalization of the benzene core like an imaginary cyclohexatriene. **d** In-silico studies based on machine learning.

with CHTs[45]. The [6 + 2]-cycloaddition involving nitroso compounds remains elusive (up to 35% yield[46,47], up to 49% ee[48]), and no detailed mechanistic studies have been reported. In the reaction with benzyne species, the [4 + 2]-cycloaddition proceeded selectively toward NCD as another valence isomeric form. Landmark precedents realized the direct modification of benzenes, such as hypothetical cyclohexatrienes, into highly functionalized six-membered carbocycles (Fig. 1c)[49,50]. The present reaction using nitroso compounds can convert unactivated benzenes to functionalized cycloheptadienes via ring expansion[51]. The pharmacological importance of these products possessing a 5,7-fused ring system[52,53] and a tricyclo(3.2.2.0)nonane structure[54] encountered in bioactive molecules and natural products also motivated us to focus on this project.

Here we describe the development of divergent cycloaddition reactions of nitroso compounds and arynes with CHT/NCD using metal-carbenes generated from ynamides. A computational mechanistic analysis was also performed to elucidate the origin of the selectivities.

## Results

### Creating and applying the regression model
Our research commenced with a comprehensive theoretical analysis of CHT/NCD and unsaturated compounds to capture the reactivity trends. In certain instances of DFT analysis, frontier orbitals were located in not-target functional groups, preventing a precise estimation of the actual orbital energy gap in the context of cycloaddition (Fig. 2a). Similarly, although previous studies examined and published the HOMO/LUMO levels of various molecules (CCCBDB)[55], practical utilization of these data to evaluate reaction progress remains elusive. This discrepancy arises from the fact that even molecular substructures containing HOMO/LUMO orbitals may not effectively engage in reactions due to such factors as little orbital overlap, steric hindrance, or aromatic stability (The present experimental result cannot be predicted using Diels-Alder Oracle.). To tackle this challenge, we employed orbital visualization techniques to specifically examine the highest occupied bond orbital (HOBO) within the conjugated olefin unit in CHT/NCD, as well as the lowest unoccupied bond orbital (LUBO) associated with the unsaturated bond of interest in the enophile. On the basis of the visual confirmation method with quantum computation, a total of 552 diverse molecules were explored to evaluate bond orbitals and

HOBO/LUBO energy levels (for details, see Supplementary Figs. 23–37).

DFT calculations and their input information provided us with a molecular dataset containing orbital information as the response variable and simplified molecular-input line-entry system strings (SMILES) as the explanatory variable for supervised machine learning[56,57]. Based on the original database constructed, diverse predictive models were created with regression techniques, including Elastic Net-, Support Vector Machine-, and Neural Network regression, and molecular descriptors, including Avalon-, Topological-, and Morgan-fingerprints with various bits (for details, see Supplementary Fig. 4). By tuning the (hyper-)parameters for a learning algorithm, approximately 650,000 predictive models were examined. Figure 2b shows the selected prediction accuracy, with cells colored in shaded hues representing accuracy, ranging from green (excellent) to red (poor). Consequently, the combination of a three hidden-layer Neural Network[58,59] and a 4096-bit Avalon fingerprint yielded a high coefficient of determination (detailed heat maps and correlation plots are available, Supplementary Figs. 5–11). Subsequently, we selected 80 compounds in the chemical information database (SciFinder[n]) and tested them using the created algorithm (Supplementary Software 1, Fig. 2c). The Neural Network prediction revealed that clusters of alkenes, alkynes, and carbonyl functionalities are less reactive as enophile components for this reaction system. Nitroso compounds and benzynes in addition to triazolines[60], however, constituted reliable candidate clusters with sufficient reactivity against CHT/NCD due to their low HOBO/LUBO gap. The use of the model provides a significant advantage because it allows for swift prediction of potential reactivities of object molecules in orbital-controlled reactions with the mere input of SMILES, thereby eliminating the laborious calculations.

### Experimental verification
Following the in-silico study, we performed in-vitro experiment using nitroso compounds and benzyne precursors (Fig. 3). The reaction of a nitrosobenzene with the valence isomers consisting of CHT (**1**) and NCD (**2**) in chlorobenzene solvent at 50 °C for 30 h (for details, see Supplementary Fig. 16) led to a CHT-selective [6 + 2]-cycloaddition reaction, producing (±)-**3a** in 72% yield as a single diastereomer (*Conditions A*). Intriguingly, no other regioisomers associated with the nitroso group nor any side-products via [4 + 2]-cycloaddition with other diene units of CHT/NCD were detected. The structure and stereochemistry of **3a** were

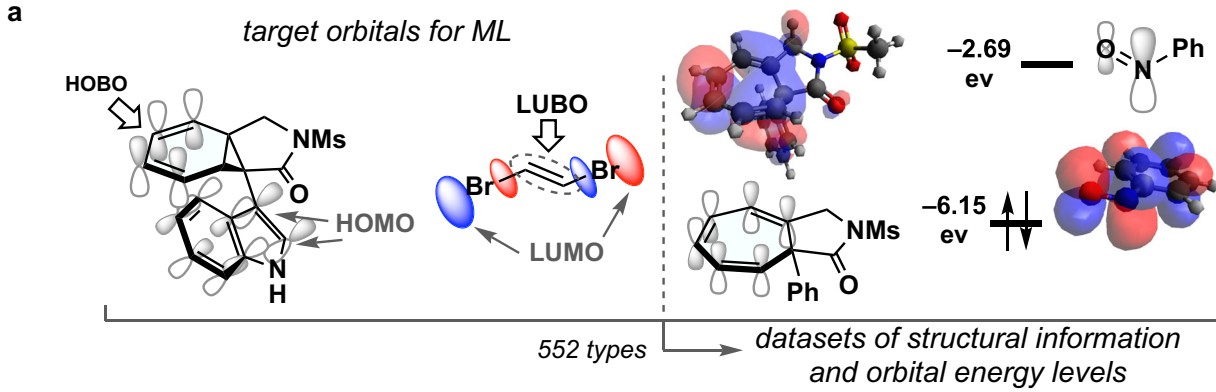

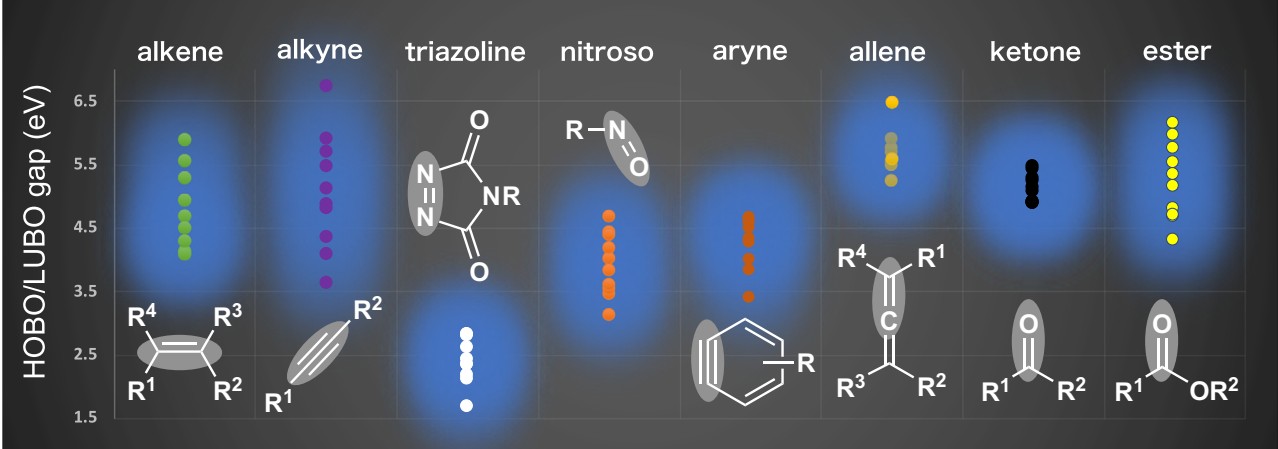

**Fig. 2 | Streamlined workflow for predicting chemical reactivity in cycloaddition.** **a** Assembly of datasets comprising orbital energy and molecular structure information using DFT computations. **b** Accuracy heat map to identify optimal algorithm combinations of diverse fingerprints and supervised machine learnings. **c** Predicted energy gap plots for orbitals dominantly involved in cycloaddition reactions. [a]radius: 4.

determined by single-crystal X-ray diffraction. The satisfactory result led us to turn our attention to the applicability of the hetero [6 + 2]-cycloaddition using nitroso compounds as a trienophile. Nitroso arenes with electron-donating and electron-withdrawing functionalities, such as -OMe, -Br, -Cl, and -CO$_2$Me, at the *ortho*-, *meta*-, and *para*-positions were usable, providing (±)-**3e**-(±)-**3j** in good yields with excellent selectivities (74%–86% yields, a single regioisomer in all cases). The reaction outcome remained largely unaffected by the electron density of the sulfonamide unit ((±)-**3b**-(±)-**3d**). The scope of CHT was illustrated by synthesizing a series of tricyclic products comprising electron-deficient or electron-abundant arenes, such as ArCN and ArOR ((±)-**3k**-(±)-**3x**).

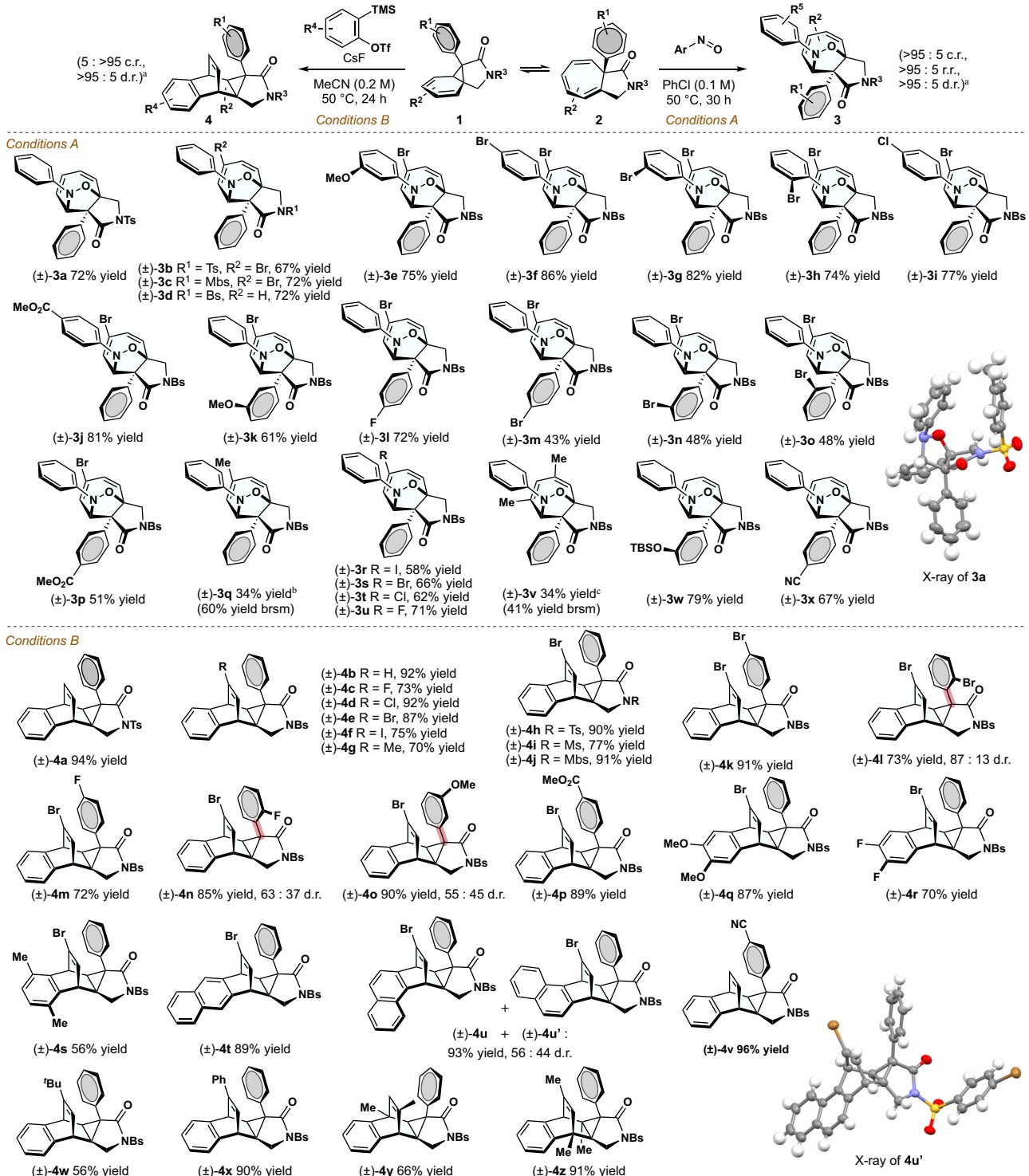

**Fig. 3 | Scope of CHT- and NCD-selective cycloaddition.** [a] > 95:5 ratio unless otherwise reported. [b] 43% of SM was recovered. [c] 17% of SM was recovered. c.r., chemoselectivity ratio; d.r., diastereomeric ratio; r.r., regioisomeric ratio; Mbs, *p*-methoxybenzenesulfonyl group; Bs, *p*-bromobenzenesulfonyl group.

Subsequently, we examined the reaction using benzyne species generated in situ from 2-(trimethylsilyl)phenyl trifluoromethanesulfonate (1.25 equiv.) with cesium fluoride (*Conditions B*). In contrast to the prior outcome, [4 + 2]-cycloaddition proceeded from minor valence-isomer (**1**) in acetonitrile solvent at 50 °C in 1 day to produce highly fused cyclohexene (±)-**4a** in 94% yield. This pericyclic reaction displayed broad substrate applicability in terms of NCD and benzyne variants without the formation of other isomers via [6 + 2]-cycloaddition ((±)-**4b**−(±)-**4j**). Pentacyclic compounds

possessing electron-poor and electron-rich aryl groups at the α-position of the amide were synthesized in good to excellent yields ((±)-**4k**−(±)-**4p**, in 72%−91%). Regarding the dienophile scope, di-fluoro and di-methoxy benzynes were applicable, delivering the corresponding product with the tricyclo(3.2.2.0)nonane structure ((±)-**4q** and (±)-**4r**). A nearly equal amount of two isomers was isolated when using naphthalyne species ((±)-**4u** and (±)-**4u'**). The reaction using a substrate with a *t*Bu group produced (±)-**4w** in a relatively lower yield than other substrate examples (56% yield).

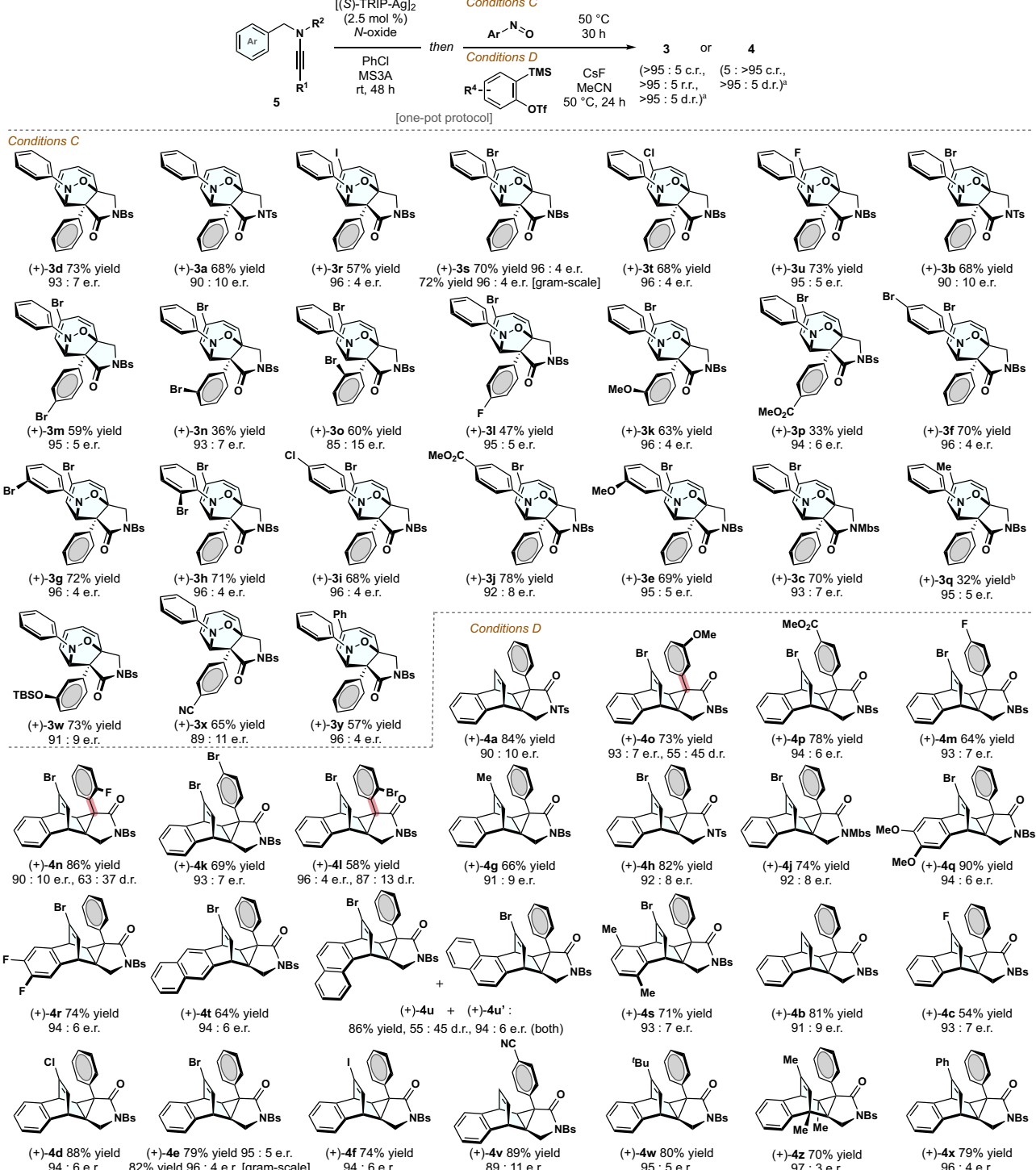

**Fig. 4 | Asymmetric synthesis of elaborate architectures via enantioselective dearomatization followed by valence-isomer selective cycloaddition.** [a] > 95:5 ratio unless otherwise reported. [b] 38% of CHT was recovered.

Thereafter, our investigation pivoted toward an asymmetric assembly of the fused cyclic molecules (Fig. 4). Chiral CHD/NCD could be synthesized via enantioselective dearomatization of non-activated arenes using silver-carbene species generated under diazo-free conditions. We envisaged that the telescoping of these reaction sequences would facilitate expeditious access to elaborate molecules with two adjacent quaternary stereogenic centers from easily available reactants. Following the reaction of ynamide **5** with 2.5 mol % of [(S)-TRIP-Ag]₂ and mild oxidant, ( + )-**3d** was obtained in an asymmetric format

through a one-pot addition of nitrosobenzene (93:7 er). That is, the selectivity preferences between the [6 + 2] and [4 + 2]-cycloaddition process were not affected by the catalyst or additives used. Substrate evaluation was then performed under the reaction protocol. High enantioselectivities were observed for an array of substrates, affording densely functionalized cycloheptadienes **3** in a stereodefined manner (*Conditions C*). The atom-economical [6 + 2]-cycloaddition could furnish cycloheptadienes with functionalities useful in coupling reactions for further derivatization ((+)-**3r**–(+)-**3u**, 57%–73% yields, 95:5–96:4 er).

Silver-catalyzed transformation using a biaryl compound furnished (+)-**3y** with a high level of enantiocontrol (96:4 er).

As presented in Fig. 4 below, a one-pot asymmetric assembly of benzo-fused ring systems was investigated using arynes. The compatibility issue between *N*-oxide with benzyne species necessitated the use of 5 equivalents of the benzyne precursor (for details, see Supplementary Fig. 18), while the residual reagents had no effect on the cycloaddition mode. *Conditions D* provided an array of penta-substituted cyclopropanes in good yields with good enantioselectivities ((+)-**4a**, (+)-**4o**, (+)-**4p**, 73%–84% yields, 90:10–94:6 er). Similarly, highly electron-rich and electron-poor benzyne species were also applicable, synthesizing (+)-**4q** and (+)-**4r** with a high level of stereocontrol (94:6 er). Intriguingly, NMR experiments revealed the presence of nonbiaryl C(sp^3)–C(sp^2) atropisomers for (+)-**4l**, (+)-**4o**, and (+)-**4n**. It is rather surprising that despite the diminutive size of the fluorine atom, each individual isomer of (+)-**4n** was isolable, indicating a unique atropisomerism in elaborate molecules. Using this strategy, we attempted to access a more complex substitution pattern and successfully synthesized (+)-**4z** with a nona-substituted cyclohexene structure. In scale-up studies, the [6 + 2]- and [4 + 2]-cycloadducts were synthesized without compromising the chemical yields and enantioselectivities ((+)-**3s** and (+)-**4e**). This streamlined procedure created four C–C bonds and a C–O double bond along with five consecutive stereocenters. Notably, substrate **5** can be synthesized in two steps from feedstock benzaldehyde or benzylamine variants, thus enabling the swift synthesis of chiral compounds **3** and **4** in a three-step sequence.

Figure 5 displays a derivatization of the products to demonstrate the utility of the present strategy. Hydrogenation of (+)-**3a** yielded a saturated cycloheptane **6** while simultaneously cleaving the N–O bond. A cyclic amide or isoxazolidine ring was selectively cleavable through LAH or Al/Zn reduction (**7** or **9**). Seven-membered amino cyclitol **12** bearing seven consecutive stereogenic centers was synthesized via sequential diastereoselective oxidation. The fully stereodefined architecture would be difficult to synthesize by alternative methods, and thus the present strategy is expected to contribute to advances in sugar chemistry. The bromoalkene moiety could be functionalized through coupling reactions (**13 → 14**), allowing for on-demand transformation.

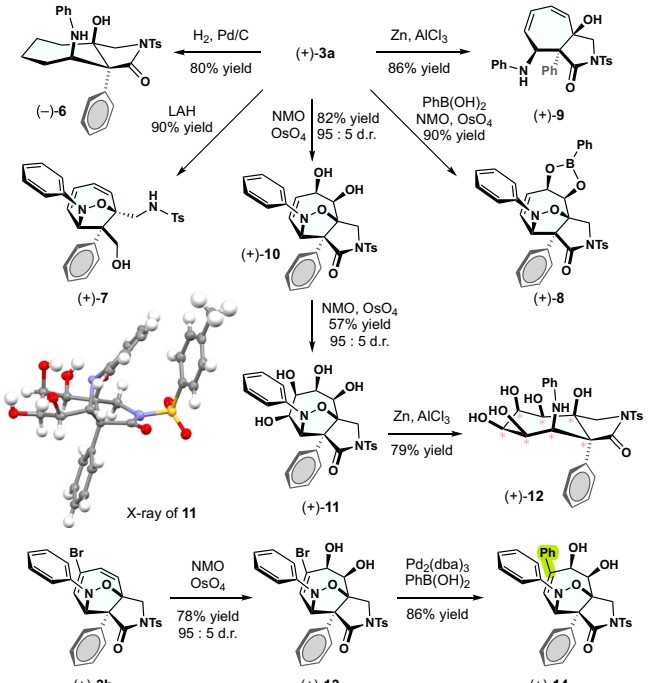

**Fig. 5 | Product derivatizations.** Synthesis of stereodefined 7-membered carbocycles through oxidative and reductive transformations.

## Computational studies

To unravel the intriguing origins behind the striking inversion of the selectivity between [6 + 2] and [4 + 2]-cycloaddition depending on the type of enophiles used, we performed a detailed mechanistic analysis. Although the developed ML-based regression technique enables the selection of reliable enophile components, the mode of cycloaddition cannot currently be predicted; hence, each reaction coordinate was analyzed by quantum chemical calculations. Firstly, the cycloaddition mode of benzyne species was computed on the basis of RωB97X-D/6-311+G** with a closed-shell singlet state (Fig. 6a). The activation energy of transition state **TS1**$_{BZ\_4+2}$ for the experimentally observed NCD-selective reaction totaled +9.39 kcal/mol according to the Curtin-Hammett principle, yielding the cycloadduct **PD**$_{BZ\_4+2}$ in a highly exothermic manner (−84.01 kcal/mol). The presence of an aryl group at the α-position of the amide obstructed the approach of the aryne from the upper side, thereby governing the π-facial stereoselectivity. Meanwhile, no single elementary reaction for the concerted suprafacial [$_π$6$_s$ + $_π$2$_s$]-cycloaddition, which is forbidden by Woodward-Hoffman rules, was detected on the potential energy surface. A postulated pathway generating an imaginary [6 + 2]-product **PD**$_{BZ\_6+2}$ involves passing through **INT3**$_{BZ\_6+2}$ and **TS3**$_{BZ\_6+2}$, but the process was not feasible with an excessively high activation energy ($\triangle\triangle G$ = +46.39 kcal/mol).

Next, nitroso cycloadditions were analyzed, including an experimentally unobserved hetero Diels-Alder reaction (Fig. 6b). An asynchronous concerted transition structure for [4 + 2]-cycloaddition was calculated to have an activation energy of +26.69 kcal/mol (**TS1**$_{NO\_4+2}$). The resulting product **PD**$_{NO\_4+2}$ has a strained structure, however, and was more thermodynamically unstable than the initial state. Indeed, StrainViz analysis indicated that **PD**$_{NO\_4+2}$ adopted a fused ring system with noticeable strain specifically around the cyclopropane[61]. On the other hand, although slower, the C–N bond formation proceeded with an activation energy of +29.33 kcal/mol from **CHT'**, generating 1,8-dipole species **INT2**$_{NO\_6+2}$. Given the relatively high activation energy, prolonging the reaction duration and elevating the temperature was likely necessary. The subsequent intramolecular charge neutralization occurred very rapidly with almost no energy barrier, furnishing **PD**$_{NO\_6+2}$ along with the formation of a C–O bond. While a distinct π–π stacking interaction, substantiated by noncovalent interaction (NCI) plotting, was observed in **NCI_TS1**$_{NO\_6+2}$, the enophile component necessitated an approach from the concave face. Conversely, the nitroso compound can approach from the convex face in the **TS1**$_{NO\_4+2}$ due to the unique valence tautomerism between the cyclic amide-fused CHT–NCD equilibrium. These influential factors would create an energy gap between **TS1**$_{NO\_4+2}$ and **TS1**$_{NO\_6+2}$ as well as **TS1**$_{BZ\_4+2}$ and **TS1**$_{BZ\_6+2}$. Although no concerted transition state[62,63] from **CHT'** to **PD**$_{NO\_6+2}$ was identified, the presence of a transit midway point, observed as a shoulder peak (**INT2**$_{NO\_6+2}$), allowed the reaction to bypass a symmetry-forbidden elementary process. **TS1**$_{NO\_6+2}$ and **TS2**$_{NO\_6+2}$ possess a remarkably alike yet distinct transition structure and conformation, empowering the higher-order [6 + 2]-cycloaddition. The computation of other site-, regio-, and periselectivities are provided in the Supplementary Fig. 3.

## Discussion

We successfully developed a neural network regression model for HOBO/LUBO gap prediction using a comprehensive dataset that encompasses the computed output of diverse molecules. Machine learning-based prediction of frontier orbital information identified benzyne species and nitroso compounds as reliable reaction components for cycloaddition processes with CHT/NCD. Based on the insights, actual experiments were performed, showing unique selectivities wherein NCD exhibited arynophile behavior while CHT exhibited nitrosophile characteristics to produce [6 + 2]-cycloadducts. By combining the asymmetric dearomatization, non-activated benzenes were converted to bridged cyclohexene in a 6/6/3/5 ring system and cycloheptadiene in a 5/7/5 ring system via ring expansion. To showcase

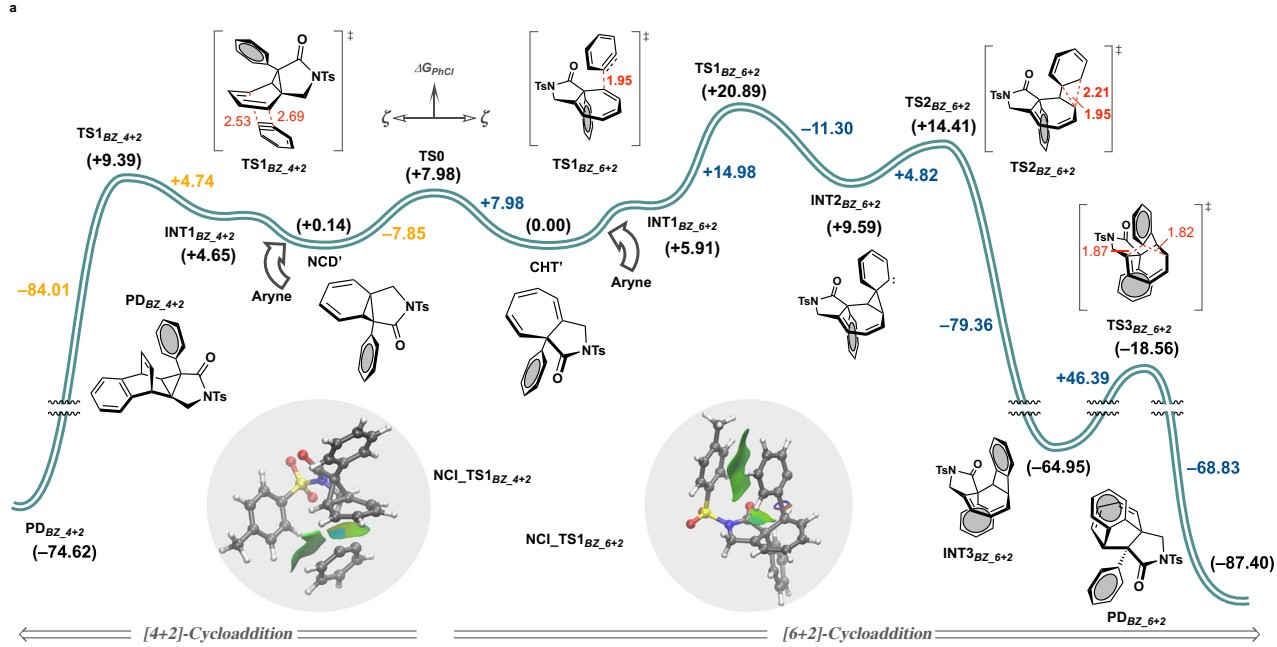

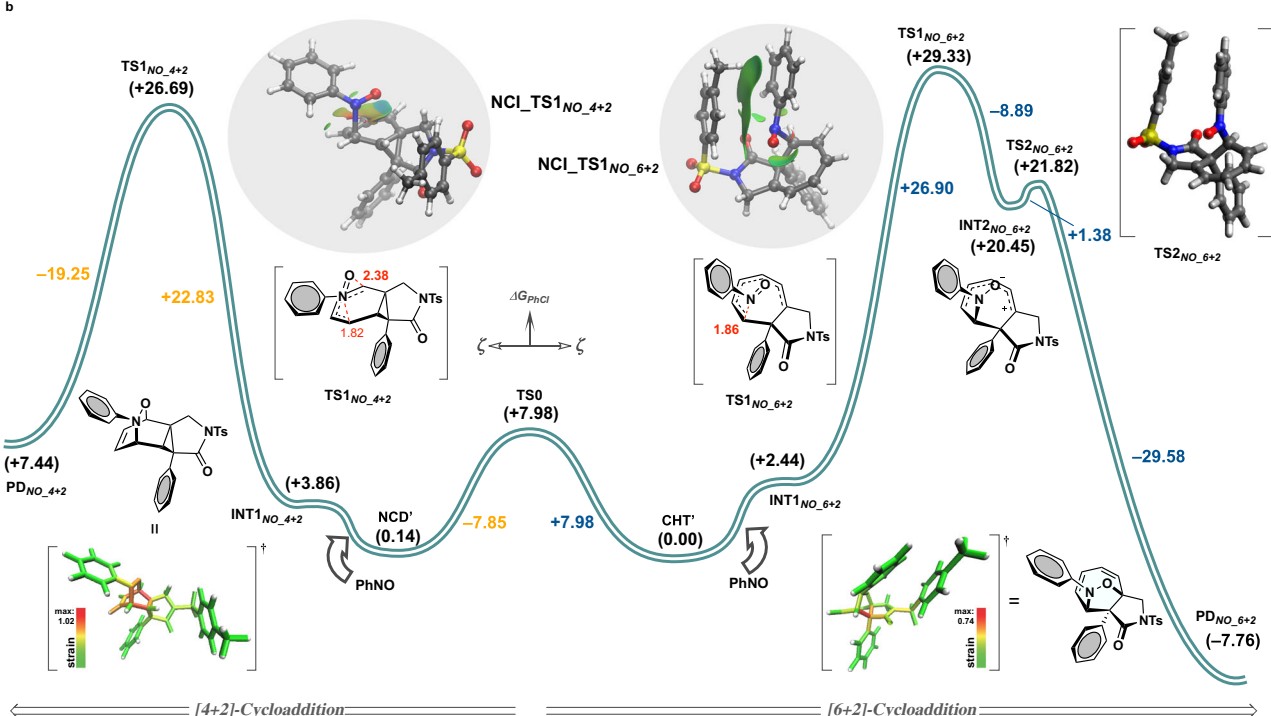

**Fig. 6 | Reaction coordinate diagram with relative Gibbs-free energies.**
**a** Benzyne-mediated pathways giving [4 + 2]-cycloadduct and hypothetical [6 + 2]-cycloadduct. **b** Nitroso compound-mediated pathways giving [6 + 2]-cycloadduct and hypothetical [4 + 2]-cycloadduct. Computation was carried out based on RωB97X-D/6-311+G**//RωB97X-D/6-31 G* in chlorobenzene solvent. The values are given in kcal/mol. Bond lengths are shown in Å. †Strained parts were colored red.

the utility of the developed method, the synthesized product was transformed into a stereodefined amino cyclitol.

## Methods

### Representative procedure for the cycloadditions with arynes (*Conditions B*)

A pre-dried 10-mL test tube equipped with a magnetic stir bar was charged with cycloheptatriene compound **2** (0.1 mmol, 1 equiv.), 2-(trimethylsilyl)phenyl trifluoromethanesulfonate (0.125 mmol, 1.25 equiv.) and CsF (0.25 mmol, 2.5 equiv.). Dry MeCN (0.5 mL, 0.2 M) was injected

into the test tube under an argon gas atmosphere. The reaction mixture was stirred for 24 h at 50 °C. The mixture was then diluted with DCM and filtered through a short pad of silica gel. After the solvent was evaporated in vacuo, the resulting residue was purified by flash chromatography (*n*-hexane/EtOAc = 5/1, v/v) to afford desired product (±)-**4**.

### Representative procedure for the cycloadditions with nitroso compounds (*Conditions C*)

A pre-dried 10-mL test tube equipped with a magnetic stir bar was covered with aluminum foil to avoid light exposure. The test tube was

charged with ynamide compound **5** (0.1 mmol, 1 equiv.), 8-methylquinoline *N*-oxide (0.2 mmol, 2 equiv.), [(S)-TRIP-Ag]₂ (0.025 mmol, 2.5 mol %) and MS3A (100 mg, 1 g/mmol), which were subsequently dissolved partially in dry PhCl (1 mL, 0.1 M) under an argon gas atmosphere. The reaction mixture was stirred for 48 h at room temperature, and then the aluminum foil was removed. Nitrosobenzene (0.3 mmol, 3 equiv.) was added to the reaction mixture, and the reaction mixture was stirred at 50 °C for 30 h. Subsequently, the reaction mixture was passed through Celite to remove MS3A. After the solvent was evaporated in *vacuo*, the resulting residue was purified by flash chromatography (*n*-hexane/EtOAc = 5/1, v/v) to afford the desired product (+)-**3**.

DFT calculations were performed with the Gaussian 16 program. The molecular structure optimizations were carried out using the RωB97X-D functional and the 6-31 G* basis set for H, C, N, O, and S. The vibrational frequencies were computed at the same level to check whether each optimized structure is at an energy minimum on the potential energy surfaces (no imaginary frequency) or a transition state (one imaginary frequency) and to evaluate its zero-point vibrational energy (ZPVE) and thermal corrections at 298.15 K. The intrinsic reaction coordinate (IRC) method was used to track minimum energy paths from transition structures to the corresponding local minima. Single point energies were calculated at the RωB97X-D level using the 6-311+G** basis set for H, C, N, O, and S in chlorobenzene solvent.

## Data availability
Experimental procedures, characterization of the compounds, and computational data are available in the Supplementary Information. Crystallographic data for the structures reported in this Article have been deposited at the Cambridge Crystallographic Data Center, under deposition numbers CCDC 2286950 (compound **3a**), 2286951 (compound **4u'**), and 2286952 (compound **11**). Copies of the data can be obtained free of charge via https://www.ccdc.cam.ac.uk/structures/. All data are available from the corresponding authors upon request. Source data are provided in this paper.

## Code availability
Program codes and datasets of machine learning are available in the Supplementary Information. The code is uploaded to GitHub (https://github.com/ShingoHaradaGit/HOBO_LUBO_Predictor).

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

## Acknowledgements

This work was supported by Digitalization-driven Transformative Organic Synthesis (Digi-TOS) from the Ministry of Education, Culture, Sports, Science & Technology, Japan (22H05337, S.H.), Takeda Science Foundation (2023077185, S.H.), Futaba Electronics Memorial Foundation (223014, S.H.), and the Sumitomo Foundation (210318, S.H.). It was also supported by JSPS KAKENHI Grant Numbers 22H02741 (T.N.), 21K06471 (S.H.), and 22KJ0470 (T.I.). Numerical calculations were carried out on SR24000 at the Institute of Management and Information Technologies, Chiba University of Japan. We thank Dr. Tomoki Yoneda (Hokkaido University) for assistance and advice regarding X-ray crystallographic analysis.

## Author contributions

S.H. and N.T. conceived and directed the project. T.I. developed the regression model and analyzed the reaction mechanism. H.T. developed the cycloaddition reactions. H.K. performed the product derivatization. H.T., T.I. and H.K. contributed to expanding the scope of the cycloaddition reactions. S.H. wrote the manuscript with contributions from all authors.

## Competing interests

The authors declare no competing interests.
