## [Peer Review File · Nature Communications]

Valence-isomer selective cycloaddition reaction of cycloheptatrienes-norcaradienesREVIEWER COMMENTS

Reviewer #1 (Remarks to the Author):

This manuscript by Harada, Nemoto, and coworkers report a synthetic strategy to afford chemo- and stereoselective cycloaddition reactions of cycloheptatrienes (CHTs) and norcaradienes (NCDs), which are valence isomers of each other in equilibrium. The authors demonstrate that the CHT isomer can undergo selective [6+2] cycloadditions with nitroso compounds, and that the NCD isomer can undergo selective [4+2] cycloadditions with benzyne. As described by the authors, the nitroso and benzyne (enophile) cycloaddition partners were chosen via a machine learning (ML) workflow that predicted the gap between the highest occupied bond orbital ("HOBO") of the conjugated olefin and the lowest unoccupied bond orbital ("LUBO") of the enophile. Reactivity of CHT and NCD then is then demonstrated in several substrate scopes, both from CHT/NCD as the starting material as well as a telescoped procedure from ynamide. Density functional theory calculations are then used to investigate the mechanism and rationalize the benzyne vs nitroso selectivity.

Overall, the chemistry presented in the manuscript is of high quality and interest. The mechanistic rationale (via DFT calculations) for the observed selectivity is also presented well. Synthetically, I think this manuscript puts forth a valuable advance in cycloaddition chemistry to afford complex structures. The derivatization reactions (Fig 3) are also a nice demonstration of the potential value of the approach in complex molecule synthesis. The substrate scopes are large with generally good yields and selectivities. However, I will note that the substrate scopes aren't all that diverse, mostly exploring installation of halide groups on the rings. Were other functional groups and/or more complex substitution patterns tried? Even if they failed, it would be good to see the full limits of the transformations with respect to productive reactivity and selectivity.

My biggest issue with this manuscript is the ML portion. The introduction of the ML modeling indicates that the "cycloaddition reactions with CHT/NCD could be designed by machine-learning algorithms". However, the ML workflow as presented does not design the reaction or conditions, nor does it predict the outcome. Instead, it provides a model to predict the "HOBO"/"LUBO" gap between CHT/NCD and the enophile from simpler molecular fingerprints (4096-bit Avalon in their case) instead of quantum mechanical (DFT) calculations. In a nutshell, all it does is reduce the computational cost of getting roughly the same HOBO/LUBO gap information that DFT gives. One note here is that I did not notice out of sample validation, which should be done or confirmed.

While generating molecular descriptors in a less computationally intensive manner is valuable, it's different from prediction of reaction conditions and/or rationalization of outcome/selectivity (see work by Aspuru-Guzik, Sigman, Doyle, and others). Furthermore, the most important synthetic and mechanistic takeaway from this paper is that CHT and NCD can be functionalized selectively from one another using different enophiles. Since the ML doesn't predict or rationalize this, it just confuses the message.

I think the ML can remain part of the manuscript, but in my opinion, it should be made crystal clear that it's being used to predict molecular properties more efficiently, not reaction conditions/reactivity. The predicted molecular property (HOBO/LUBO gap) may be correlated with reactivity, but this is still distinct. To this end, as far as I can tell, the authors also did not demonstrate that their HOBO/LUBO model even is correlated with successful/unsuccessful reactivity. This further muddles the point of the ML section in my

opinion, and should be addressed.

Reviewer #2 (Remarks to the Author):

The authors reported a neural network regression model using bond orbital data to predict chemical reactivities. The results were experimentally confirmed with cycloheptatriene-selective (6 + 2)-cycloaddition utilizing nitroso compounds and norcaradiene-selective (4 + 2)-cycloaddition reactions employing benzyne. This work provides a rational explanation for the seemingly anomalous occurrence of thermally prohibited suprafacial (6 + 2)-cycloaddition. These are the noteworthy results.

This work is original and will be of significance to AI applied in Chemistry. The conclusions are supported by the data reported.

The methodology is sound, meets the expected standards in Chemistry.

Reviewer #3 (Remarks to the Author):

The paper titled "Valence-isomer selective cycloaddition reaction of cycloheptatrienes/norcaradienes" by Nemoto et al presents a remarkable contribution to the field of synthetic chemistry by combining data science and organic synthesis. The authors successfully developed an artificial neural network regression model using bond orbital data to predict chemical reactivities, which was experimentally verified through cycloaddition reactions utilizing nitroso compounds and benzyne.

The integration of machine learning techniques with organic synthesis is an emerging trend, and this study provides a compelling example of its potential. The use of computational studies to explain the seemingly anomalous occurrence of thermally prohibited suprafacial (6 + 2)-cycloaddition without photoirradiation adds further depth to the research.

One of the notable achievements of this work is the one-pot asymmetric synthesis achieved by telescoping the enantioselective dearomatization of non-activated benzenes and cycloadditions. This approach allows for the rapid and efficient creation of complex molecules with control over multiple selectivities. The successful synthesis of penta-substituted cyclopropanes with good yields and enantioselectivities demonstrates the practicality and potential applications of this methodology.

To further elevate the manuscript, the authors could consider addressing the following questions:

1. How does the neural network regression model compare to other predictive models in terms of accuracy and efficiency? What are the potential advantages and disadvantages of using an artificial neural network regression model for reaction design and synthesis planning?
2. Can the developed model be applied to predict reactivities and selectivities in other types of cycloaddition reactions?

3. What are the limitations and challenges in scaling up the one-pot asymmetric synthesis for industrial applications?

4. Are there any potential applications or implications of this methodology in drug discovery or materials science?

5. Could you provide more details on the specific bond orbital data used in the artificial neural network regression model? How were these data obtained and validated?

6. In the computational studies, what were the key factors or parameters that allowed for the occurrence of the thermally prohibited suprafacial (6 + 2)-cycloaddition without photoirradiation? Can you provide a more detailed explanation of the underlying mechanism?

Overall, this paper contributes to the field of synthetic chemistry and showcases the potential of data-driven approaches. Considering its scientific significance and innovative methodology, this paper would be worthy of publishing in Nature Communications journal.

Point by point response to the reviewers' comments

Manuscript ID: NCOMMS-23-49323

Title: Valence-isomer selective cycloaddition reaction of cycloheptatrienes-norcaradienes

Authors: Shingo Harada*, Hiroki Takenaka, Tsubasa Ito, Haruki Kanda, Tetsuhiro Nemoto*

Dear Reviewers

We would like to thank the referees for spending time on this paper and providing insightful comments. According to the comments from the editor and referees, we have carefully revised our manuscript. The revised points are highlighted in yellow. The detailed revisions were listed as follows:

Reviewer 1

General comment:

This manuscript by Harada, Nemoto, and coworkers report a synthetic strategy to afford chemo- and stereoselective cycloaddition reactions of cycloheptatrienes (CHTs) and norcaradienes (NCDs), which are valence isomers of each other in equilibrium. The authors demonstrate that the CHT isomer can undergo selective [6+2] cycloadditions with nitroso compounds, and that the NCD isomer can undergo selective [4+2] cycloadditions with benzyne. As described by the authors, the nitroso and benzyne (enophile) cycloaddition partners were chosen via a machine learning (ML) workflow that predicted the gap between the highest occupied bond orbital (“HOBO”) of the conjugated olefin and the lowest unoccupied bond orbital (“LUBO”) of the enophile. Reactivity of CHT and NCD then is then demonstrated in several substrate scopes, both from CHT/NCD as the starting material as well as a telescoped procedure from ynamide. Density functional theory calculations are then used to investigate the mechanism and rationalize the benzyne vs nitroso selectivity.

Overall, the chemistry presented in the manuscript is of high quality and interest. The mechanistic rationale (via DFT calculations) for the observed selectivity is also presented well. Synthetically, I think this manuscript puts forth a valuable advance in cycloaddition chemistry to afford complex structures. The derivatization reactions (Fig 3) are also a nice demonstration of the potential value of the approach in complex molecule synthesis.

Comment 1:

The substrate scopes are large with generally good yields and selectivities. However, I will note that the substrate scopes aren't all that diverse, mostly exploring installation of halide groups on the rings. Were other functional groups and/or more complex substitution patterns tried? Even if they failed, it would be good to see the full limits of the transformations with respect to productive reactivity and selectivity.

Response 1:

We are thankful for the reviewer's constructive comments. Based on the Comment 1 from Reviewer 1, we performed the additional investigation on the substrate scope in the cycloaddition reactions using nitroso compounds and benzyne species to synthesize the corresponding racemic and chiral products. The results are summarized below. Successful examples have been added to the manuscript, and unsuccessful examples have been added to the Supplementary information file (Table S13). The sentences added to the manuscript and the revised sentences are as follows.

About the *Conditions A*, "The scope of CHT was illustrated by synthesizing a series of tricyclic products comprising electron-deficient or electron-abundant arenes, such as ArCN and ArOR ((±)-**3k**–(±)-**3x**)". About the *Conditions B*, "Reaction using a substrate with a 'Bu group produced (±)-**4w** in a relatively lower yield than other substrate examples (56% yield)". About the *Conditions C*, "Silver-catalyzed transformation using a biaryl compound furnished (+)-**3y** with a high level of enantiocontrol (96 : 4 er)". About the *Conditions D*, "Using this strategy, we attempted to access a more complex substitution pattern and successfully synthesized (+)-**4z** with a nona-substituted cyclohexene structure".

successful examples

Compound data, HPLC charts, and NMR charts (including substrate synthesis) have been added to the Supplementary information file.

unsuccessful examples

Comment 2:

My biggest issue with this manuscript is the ML portion. The introduction of the ML modeling indicates that the “cycloaddition reactions with CHT/NCD could be designed by machine-learning algorithms”. However, the ML workflow as presented does not design the reaction or conditions, nor does it predict the outcome. Instead, it provides a model to predict the “HOMO”/“LUBO” gap between CHT/NCD and the enophile from simpler molecular fingerprints (4096-bit Avalon in their case) instead of quantum mechanical (DFT) calculations. In a nutshell, all it does is reduce the computational cost of getting roughly the same HOMO/LUBO gap information that DFT gives. One note here is that I did not notice out of sample validation, which should be done or confirmed.

While generating molecular descriptors in a less computationally intensive manner is valuable, it's different from prediction of reaction conditions and/or rationalization of outcome/selectivity (see work by Aspuru-Guzik, Sigman, Doyle, and others). Furthermore, the most important synthetic and mechanistic takeaway from this paper is that CHT and NCD can be functionalized selectively from one another using different enophiles. Since the ML doesn't predict or rationalize this, it just confuses the message.

Response 2:

Thank you very much for the thoughtful comments. Regarding the sentence for the working hypothesis in the introduction section, we revised the original sentence to the following. “Our working hypothesis is that machine-learning algorithms of the frontier orbital energy gaps predicted by computations based on density functional theory (DFT) could find enophile candidates that react efficiently with CHT/NCD.” Besides, we uploaded the program of the

sample validation (EP_LUBO.pkl, CHTNCD_HOBO.pkl, Orbital_Energy_Level_Predictor.py, README.md files. etc). By using these files, all the user has to do is draw the structure in ChemDraw, and copy and paste it as SMILES. Thus, molecular properties can be quickly predicted using the user interface without performing DFT calculations or writing programming code. We believe that this is important because even some talented scientists do not have access to expensive hardware/software to perform DFT calculations due to economic reasons.

Comment 3:

I think the ML can remain part of the manuscript, but in my opinion, it should be made crystal clear that it's being used to predict molecular properties more efficiently, not reaction conditions/reactivity. The predicted molecular property (HOBO/LUBO gap) may be correlated with reactivity, but this is still distinct. To this end, as far as I can tell, the authors also did not demonstrate that their HOBO/LUBO model even is correlated with successful/unsuccessful reactivity. This further muddles the point of the ML section in my opinion, and should be addressed.

Response 3:

Based on the comments, we revised the phrase from “*Reaction Design*” to “*Prediction of Molecular Properties*” in Fig 1 (d).

Additionally, we performed some additional investigations. The results are summarized below in the figures, showcasing the comparison between machine-learning prediction and the empirical results. Although nitrosobenzene, aryne, and triazoline caused orbital-

controlled reactions with CHT/NCD, other functionalities did not show sufficient reactivity. Consequently, the HOBOLUBO regression model is generally correlated with successful/unsuccessful reactivity. These results were added to the Supplementary information file (Table S14).

Reviewer 2

All Comments:

The authors reported a neural network regression model using bond orbital data to predict chemical reactivities. The results were experimentally confirmed with cycloheptatriene-selective (6 + 2)-cycloaddition utilizing nitroso compounds and norcaradiene-selective (4 + 2)-cycloaddition reactions employing benzyne. This work provides a rational explanation for the seemingly anomalous occurrence of thermally prohibited suprafacial (6 + 2)-cycloaddition. These are the noteworthy results. This work is original and will be of significance to AI applied in Chemistry. The conclusions are supported by the data reported. The methodology is sound, meets the expected standards in Chemistry.

Response:

We sincerely appreciate the comments that accurately assess the strengths of this research project. Thanks again for considering our work.

Reviewer 3**General comment:**

The paper titled "Valence-isomer selective cycloaddition reaction of cycloheptatrienesnorcaradienes" by Nemoto et al presents a remarkable contribution to the field of synthetic chemistry by combining data science and organic synthesis. The authors successfully developed an artificial neural network regression model using bond orbital data to predict chemical reactivities, which was experimentally verified through cycloaddition reactions utilizing nitroso compounds and benzyne.

The integration of machine learning techniques with organic synthesis is an emerging trend, and this study provides a compelling example of its potential. The use of computational studies to explain the seemingly anomalous occurrence of thermally prohibited suprafacial (6 + 2)-cycloaddition without photoirradiation adds further depth to the research.

One of the notable achievements of this work is the one-pot asymmetric synthesis achieved by telescoping the enantioselective dearomatization of non-activated benzenes and cycloadditions. This approach allows for the rapid and efficient creation of complex molecules with control over multiple selectivities. The successful synthesis of penta-substituted cyclopropanes with good yields and enantioselectivities demonstrates the practicality and potential applications of this methodology.

To further elevate the manuscript, the authors could consider addressing the following questions:

Comment 1:

How does the neural network regression model compare to other predictive models in terms of accuracy and efficiency? What are the potential advantages and disadvantages of using an artificial neural network regression model for reaction design and synthesis planning?

Response 1:

Thank you very much for the comments. Artificial neural network regression models can understand non-linear relationships in a dataset by changing the number of nodes and activation functions. Due to its nonlinearity, it is suitable for tasks where other linear regression models may not work well. Regarding the potential disadvantages of using an artificial neural network model for synthesis planning, training neural networks may require

different data sets. Still, these datasets may not be readily available for specific reactions or reaction conditions, especially for less-studied transformations. Another concern about neural networks would be the need for more interpretability of hidden layer weight matrices. In the discipline of chemistry, since understanding the rationale behind predictions is important, the nature of the black box could be a drawback. A potential advantage of using neural networks is that, in some cases, they can successfully predict reaction outcomes with small datasets for reaction design, significantly saving time costs and synthesis effort in actual experiments. The model developed in this study requires only highly simplified input data (SMILES). Therefore, it can only be run using the ChemDraw application, which is widely available for chemists worldwide.

Comment 2:

Can the developed model be applied to predict reactivities and selectivities in other types of cycloaddition reactions?

Response 2:

The reactivities of enophile components in other types of cycloadditions would be predicted by the regression analysis of LUBO involved in the reactions. However, the current neural network regression model cannot predict the selectivity outcome. Introducing additional training data targeted at selectivity could make it achievable.

Comment 3:

What are the limitations and challenges in scaling up the one-pot asymmetric synthesis for industrial applications?

Response 3:

Based on the comment 3, we performed the reactions on a large scale. As shown in the figure below, the (6 + 2)- and (4 + 2)-cycloadducts were synthesized without compromising the chemical yields and enantioselectivities. However, the following problems may arise for industrial applications. Reaction times are long, and chiral catalyst precursors are relatively expensive. These may be remedied by a little more heating or higher demands on the catalyst. The result of gram-scale reactions has been added to the Supplementary information file.

Comment 4:

Are there any potential applications or implications of this methodology in drug discovery or materials science?

Response 4:

The products in the (4 + 2)-cycloaddition include 6/6/3-fused ring system. The characteristic structures are found in bioactive molecules such as the *anti*-orthopoxvirus compound ST-246. The derivatives of the compounds are highly effective in inhibiting HIV-1 R5 infection ((a) Dong, M.-X., Zhang, J., Peng, X.-Q., Lu, H., Yun, L.-H., Jiang, S., Dai, Q.-Y. Tricyclononene Carboxamide Derivatives as Novel anti-HIV-1 Agents. *Eur. J. Med. Chem.* **45**, 4096–4103 (2010). (b) Jordan, R., Tien, D., Bolken, T. C., Jones, K. F., Tyavanagimatt, S. R., Strasser, J., Frimm, A., Corrado, M. L., Strome, P. G., Hruby, D. E. Single-Dose Safety and Pharmacokinetics of ST-246, a Novel Orthopoxvirus Egress Inhibitor. *Antimicrob. Agents Chemother.* **52**, 1721–1727 (2008).). Additionally, the product in the (6 + 2)-cycloaddition could be transformed into all-*cis* amino cyclitol with seven consecutive stereogenic centers (Fig 3). Creating the fully stereodefined structure would be a challenging task using alternative strategies. Hence, this methodology has the potential to contribute to advances in glycochemistry or drug discovery.

Comment 5:

Could you provide more details on the specific bond orbital data used in the artificial neural network regression model? How were these data obtained and validated?

Response 5:

The specific bond orbital data were obtained using the following steps.

1) Density functional theory calculations were performed with Gaussian 16 program to optimize the molecular structure and generate comprehensive bond orbital information. The vibrational frequencies were computed to check whether each optimized structure is at an

energy minimum on the potential energy surfaces (no imaginary frequency).

2) By using “formchk” command in a terminal, checkpoint files (.chk) were converted into Gaussian formatted checkpoint file format (.fchk) with the program.

3) The obtained FCHK files were opened in Avogadro software. <https://avogadro.cc/install/>

4) When the obtained FCHK files are opened, Avogadro automatically opens the orbitals toolbar containing all potential molecular orbitals.

5) By visualizing the orbitals, we identified the occupied orbital at the highest energy level of the functional groups of interest (HOBO), rather than the entire molecule.

Similarly, the unoccupied orbital at the lowest energy level of the functional groups of interest was also identified (LUBO), and the energy level information was obtained.

6) Molecular structures were drawn in ChemDraw and converted to molecular fingerprint *via* SMILES. The fingerprint was used as the independent variable and HOBO/LUBO was used as the dependent variable.

7) (1)-(6) were repeated to obtain a dataset containing about 500 molecules.

8) The dataset was divided into training data, validation data, and test data. By using the training data and the validation data, regression models were constructed and the hyperparameters were adjusted.

9) Algorithm combinations of fingerprints and supervised machine learnings were examined using the test data. As a result, the combination of a three hidden-layer Neural Network, and a 4096-bit Avalon fingerprint yielded a high coefficient of determination. (Approximately 650,000 predictive models were examined; please see sections 2 and 11 in the Supplementary information file for the details.)

10) The accuracy of the bond orbital data was validated by other data not included in training dataset.

(appendix) The following steps allow us to predict the HOBO/LUBO gap quickly and easily.

1. Please draw the structure in ChemDraw.

2. Please copy & paste as SMILES.

3. Please push the "Predict !" button.

Comment 6:

In the computational studies, what were the key factors or parameters that allowed for the occurrence of the thermally prohibited suprafacial (6 + 2)-cycloaddition without photoirradiation? Can you provide a more detailed explanation of the underlying mechanism?

Response 6:

Thank you very much for these comments. In fact, the activation energies of the (4 + 2)-cycloaddition reaction is lower than that of the (6 + 2)-cycloaddition reaction. Thus, in both reactions involving nitroso compounds and benzyne species, the (4 + 2)-cycloaddition proceeds faster than the (6 + 2)-cycloaddition (Please see also Fig. 4). However, product $\text{PD}_{\text{NO}_4+2}$ was unstable because the fused ring system including the three-membered ring is distorted. The total strain energy of $\text{PD}_{\text{NO}_4+2}$, including bond strain, angle strain, and dihedral strain, is 12.92 kcal/mol, whereas the total strain energy of $\text{PD}_{\text{NO}_6+2}$ was 6.31 kcal/mol based on the StrainViz analysis. Thereby, $\text{PD}_{\text{NO}_4+2}$ has higher energy on the potential energy surface than the initial state (SM), indicating the reversibility of the (4 + 2)-cycloaddition process.

Repeated computational searches to find the (6 + 2)-cycloaddition process did not reveal a concerted pathway. However, after N–O bond formation *via* **TS1_{NO_6+2}**, the presence of a small local minimum (**INT2_{NO_6+2}**) and an almost barrier-free transition state ($\Delta G^\ddagger = +1.38$ kcal/mol) allowed the reaction pathway to avoid symmetry-forbidden (6 + 2)-cycloaddition processes. This means that **PD_{NO_6+2}** could be reached on the potential energy surface *via* stepwise N–O/C–O bond formation. The above is a detailed explanation for the seemingly anomalous occurrence of thermally prohibited suprafacial (6 + 2)-cycloaddition without photoirradiation.

Comment 7:

Overall, this paper contributes to the field of synthetic chemistry and showcases the potential of data-driven approaches. Considering its scientific significance and innovative methodology, this paper would be worthy of publishing in Nature Communications journal.

Response 7:

Thank you very much for your six constructive questions. We wholeheartedly agree that these discussions will be very informative for broad readers.

Consequently, the revision pointed out by the reviewer made the manuscript better. We sincerely appreciate these comments.

Thank you again for your kind attention.

With best regards,

Tetsuhiro Nemoto, PhD.

Professor

Graduate School of Pharmaceutical Sciences, Chiba University

REVIEWERS' COMMENTS

Reviewer #1 (Remarks to the Author):

All of my comments and concerns have been addressed. I am in support of publication.

Reviewer #3 (Remarks to the Author):

I think the authors addressed the comments correctly and in huge depth, hence I do not have additional questions and I suggest this manuscript be published as it is without further amendments.